# Filtered beauty in Oslo and Tokyo: A spatial frequency analysis of facial attractiveness

**Morten Øvervoll[1], Ilaria Schettino[2], Hikaru Suzuki[3], Matia Okubo[3], Bruno Laeng [2,4] \***

**1** Department of Psychology, University of Tromsø (The Arctic University of Norway), Tromsø, Norway, **2** Department of Psychology, University of Oslo, Oslo, Norway, **3** Department of Psychology, Senshu University, Tokyo, Japan, **4** RITMO Centre for Interdisciplinary Studies of Rhythm, Time and Motion, University of Oslo, Oslo, Norway

* bruno.laeng@psykologi.uio.no

**Citation:** Øvervoll M, Schettino I, Suzuki H, Okubo M, Laeng B (2020) Filtered beauty in Oslo and Tokyo: A spatial frequency analysis of facial attractiveness. PLoS ONE 15(1): e0227513. https://doi.org/10.1371/journal.pone.0227513

**Data Availability Statement:** All datafiles are available from the Figshare database (accession number(s) 10.6084/m9.figshare.8311274, 10.6084/m9.figshare.8246864, 10.6084/m9.figshare.8311256, 10.6084/m9.figshare.8262497.

## Abstract

Images of European female and male faces were digitally processed to generate spatial frequency (SF) filtered images containing only a narrow band of visual information within the Fourier spectrum. The original unfiltered images and four SF filtered images (low, medium-low, medium-high and high) were then paired in trials that kept constant SF band and face gender and participants made a forced-choice decision about the more attractive among the two faces. In this way, we aimed at identifying those specific SF bands where forced-choice preferences corresponded best to forced-choice judgements made when viewing the natural, broadband, facial images. We found that aesthetic preferences dissociated across SFs and face gender, but similarly for participants from Asia (Japan) and Europe (Norway). Specifically, preferences when viewing SF filtered images were best related to the preference with the broadband face images when viewing the highest filtering band for the female faces (about 48–77 cycles per face). In contrast, for the male faces, the medium-low SF band (about 11–19 cpf) related best to choices made with the natural facial images. Eye tracking provided converging evidence for the above, gender-related, SF dissociations. We suggest greater aesthetic relevance of the mobile and communicative parts for the female face and, conversely, of the rigid, structural, parts for the male face for facial aesthetics.

## Introduction

We often perceive other people's faces as beautiful. Although strongly debated, there are elements of our sense of facial beauty that seems to be based on biological grounds (e.g., as a cue to mate value; [1–3]) or as universal learning experiences which may therefore be expressed cross-culturally as 'standards' of beauty [4–7]. However, contextual differences in judgements of attractiveness have been documented (e.g., [8–10]) and judgements of beauty can clearly differ at the individual level [11] also in part on the basis of learning of parental or self's facial characteristics [12–16]. Thus, any single observer's sense of beauty may be best thought as the result of a delicate balance between preferences acquired during the history of the species as well as the history of a specific individual [11–17].

**Funding:** The author(s) received no specific funding for this work.

**Competing interests:** The authors have declared that no competing interests exist.

A lingering question for research on facial aesthetics is what kind of visual information underlies facial attractiveness decisions (e.g., [18]). Several studies have attempted to answer this question by focusing on facial dimensions or properties like symmetry, averageness, youthfulness, femininity and masculinity (for reviews see [19–21]). However, the spatial frequency structure of visual stimulus is a fundamental property of visual information exploited by our brains. Indeed, visual areas appear to process SF information [22, 23] and separate to some extent SF information along neural streams at a very early stage (e.g., the magnocellular and parvocellular channels; e.g. [24]). Neuroimaging studies with humans indicate a brain's organization for face stimuli in separate neural areas, these based on specific bands or ranges of spatial frequency information [25], so that occipito-temporal cortex extract distinct visual cues at different SF ranges in faces and these outputs are then projected forward to the fusiform gyrus, where these different visual cues may converge. It seems plausible that several of the higher-level facial dimensions listed above (e.g., symmetry, masculinity) may depend on low-level coding of different visual spatial frequencies.

The application of Fourier theory within psychology and neuroscience to visual information processing has revolutionized theoretical thinking, at least since the previous century, by allowing the encoding of stimulus patterns as composed of multiple levels of spatial frequency (SF). Within Fourier theory, 'SF' indicates the rapidity of 'spatial change' in levels of lightness (see Fig 1 for examples of band-passed or filtered images of faces). In general, low SF-passed (LSF) images carry more global and coarse information. When viewed alone these may reveal the global shape's spatial aspects of a face, possibly better than high SF-passed (HSF) images. LSF facial images may reveal the rigid, structural, aspects of the face (e.g., proportions of the underlying skeletal structure), while preserving information about the global reflectance properties of a face and its shape's symmetry, although at low levels of filtering the coarseness of the image may also reduce the symmetry detection ability [26]. Indeed, a variety of research on either face recognition or the perception of emotional expressions focused on the role of spatial frequencies [27–37], but surprisingly very few studies have specifically investigated whether specific spatial frequency information may also underlie our sense of facial beauty. To our knowledge—no study has directly explored or compared the roles of several spatial frequency bands on aesthetic judgements. In fact, most studies on the role spatial frequencies in face processing have focused on a simple binary distinction of low-pass versus high-pass frequencies that might oversee relevant distinctions. A detailed spatial frequency information approach would seem relevant for identifying relevant aesthetic cues within faces. This may also provide converging evidence to that garnered with different methods (e.g., judgements of facial morphs) and, perhaps, suggest novel dimensions. Exploring whether specific SF bands affect in a gender-specific manner the judgements of beauty may also throw light on sex-dimorphic cues to beauty, since these may not be equally visible at all levels of spatial frequency.

Some previous research suggests that low-level image properties may provide key properties for aesthetic decisions on faces, since observers rate face images with shallower slopes of the Fourier power spectrum as more attractive compared to the Fourier slopes of the original face or the same face with a steeper slope [38]. Shallow Fourier slopes imply increased high spatial frequency (HSF), whereas steeper ones imply the opposite enhancement in low spatial frequency (LSF) power. The coarse 'low spatial frequency' (LSF) information may already contain key information about face beauty, since when the finer, high spatial frequency (HSF), information is removed or interfered with, it is still possible to make judgments of attractiveness and these seem consistent with those made when viewing intact (broadband) images. Bachmann [39] 'quantised' faces (thus degrading HSF information) and found that very coarse visual information (i.e., only 17 pixels for each face image) still supported the ability to evaluate

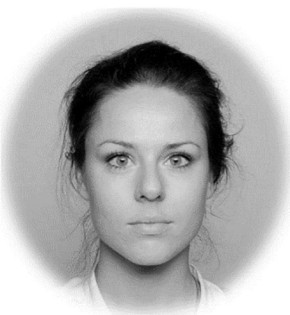

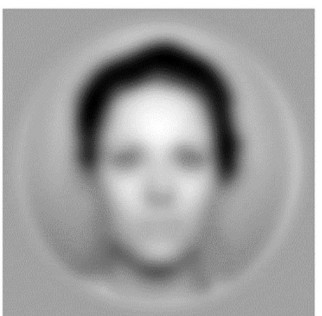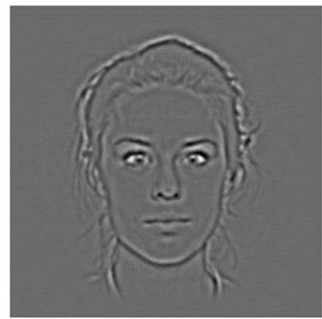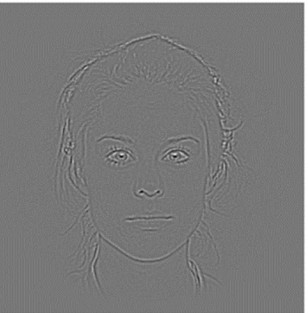

**Fig 1. Illustration of the four filtered versions of one of the female faces from the Oslo Face Database.** The top image shows the broadband (unfiltered) photo image. Bottom; from left: low-pass filtered face (0.45–4 cpf), medium-low-pass filtered face (11.13–18.76 cpf), medium-high-pass filtered face (25.83–40.5 cpf) and high-pass filtered face (47.65–77.04cpf). Nota Bene: the size of the facial stimuli in this illustration are smaller and with lower resolution than the original photo images as well as the images presented on screen during the experiments.

attractiveness. In another study where participants judged relative attractiveness of a face pair presented at several eccentricities from the central fixation [40], the discrimination performance at parafovea was indistinguishable from the performance around the fovea. Attractive faces were detected even at a relatively more peripheral location (i.e., 10˚ of eccentricity from fixation), though the ability to perceive attractiveness for the same faces decreased (but still better than chance). Both beauty and cuteness appear to be detectable in peripheral vision, though the ability to judge cuteness may decline more in peripheral vision than for judging face beauty [41]. Thus, these few studies suggest that LSF information may be sufficient to appraise attractiveness and that important cues may be contained within the lower spatial frequencies and not confined to a specific higher SF band. Interestingly, similarly to the perception of non-aesthetic emotions (e.g. [42]), attractiveness might still be perceived after extreme LSF filtering in which the perception of face identity is compromised, since face recognition drops rapidly as a function of eccentricity [43–44]. Developmental studies (e.g., [45, 46]) have shown that newborns prefer attractive faces, which implies that an immature visual system, devoid of visual experience, can process attractiveness already at birth. Considering also that the immature visual system of newborns is restricted to a range of LSF visible to adults [47], these findings also hint to a role of an early, subcortical, processing route for face aesthetics in humans [48].

The roles of LSF versus HSF information in aesthetic judgments may turn out to be quite different from those they play in the recognition of either identity or facial emotional expressions. Face identification may be mediated by a limited band of mid spatial frequencies [49–52]. In contrast, the recognition of facial emotional expressions appears to be impaired at a higher resolution threshold [53] than that necessary of aesthetic judgments [39]. The facial expressions of happiness and surprise are visible at several low-frequency bands and

recognizable from a far distance [54], whereas both high and low frequencies are important for recognition of joy and anger, with a slight preference for the high frequencies.

Hence, the present study focusses on the roles of SF in facial aesthetics and presents a way to assess the relative "weight" of specific bands of frequency information in beauty decisions. By obtaining separate attractiveness decisions with broadband images of faces (i.e., unfiltered face images) and comparing these to attractiveness decisions made by the same individuals with several filtered or band-passed versions of the same faces, it should be possible to identify which SF bands contain the most relevant cues for beauty decisions. For instance, if low spatial frequency information correlates highly (perhaps best) with broadband aesthetic choices than higher SFs, this may add evidence for attractiveness been based on perceiving coarse, global, features (e.g., the general proportions of the face or wide facial shape properties like jaw shape and size). In contrast, if high spatial frequency best correlates with a same individual's 'broadband' aesthetic choices, one could conclude for local features (e.g., shape of the eyes or nose) having most weight for attractiveness (perhaps also based on distance from averageness of a face's shape or on sexually dimorphic features that differ locally, e.g. nose, mouth). Differently from LSF-passed faces, HSF images can reveal clearly the face's textural details, fine elements of shape of the face surface (e.g., eyes, mouth, facial hair; e.g., [28], small parts' local reflectance and color, and fine creases and blemishes on the skin. Hence, whenever choices in any band of SF filtering overlap with broadband aesthetic choices, we would conclude that its SF structure includes some key aesthetic information. Further, the present 'band-pass method' may help to narrow down the search for properties that carry weight for gender-specific or culture-specific judgments of attractiveness.

Informational manipulations of stimuli (e.g., filtering) are a fundamental way of discovering which aspects of information are relevant for a particular judgement, but without additional evidence it can remain unclear which specific aspect in the stimulus an observer is actually attending. As suggested in a paper by Cronbach and Meehl [55], using multiple measures can establish optimally the measurement of a theoretical construct, allowing the disambiguation of scrutiny of alternative accounts. Eye tracking is a particularly valuable type of convergent measure for studies of cognition, since spontaneous eye scanning provide a very close estimate of the locus of attention at each moment in time and typifies the way visual attention is normally deployed [56–58]. Hence, we monitored eye fixations during a "beauty contest" task. In each trial, we showed side by side a pair of faces and requested participants to decide which of the two individuals looked more attractive, by pressing one of two keys, since previous eye-tracking studies of aesthetic decisions or consumers' preferences have shown a tight relation between gaze onto a particular item and preference [59–61]. Comparing oculomotor behavior could reveal sex differences in attending to female versus male faces [62], which again might turn out to be different for different SF bands.

Another question is whether individuals from different cultures may differ on aesthetic judgments of faces [63]. It is reasonable to assume that individuals within one culture may have differential exposure to the ethnicity of the faces of another culture. An observer may scrutinize faces of other ethnicities by applying criteria used for faces of one's own ethnicity. The shapes of faces may differ on average morphology, proportions, and/or variance across ethnicities [64–68]. Although for several types of perceptual decisions, there are seemingly no differences across ethnicities [69], European and Asians may sample facial information differently [70]. Some studies suggested that Asians are more "tuned" relatively to Westerners towards the LSF information in face stimuli (e.g., by attending holistically or utilizing a broader spread of attention [71]).

Hence, in the present study, we presented the same photo pairs of female or male European faces to a group of young European participants (in Oslo, Norway) and a group of young

Asian participants (in Tokyo, Japan) of either sex (females and males). The face pairs used as stimuli in our paradigm came in five versions, namely: low-pass filtered face pairs, medium-low-pass filtered, medium-high-pass filtered, and high-pass filtered as well as broadband (unfiltered). Based on previous studies, we expected that all band-pass images would relate above chance to attractiveness decisions made on the unfiltered or natural face images. Yet, we expected that aesthetics judgments would differ depending on the SF ranges and especially so when considering the gender of the faces and of the participants as well as the latter's ethnicity.

## Materials and methods

### Participants

There were fifty-two participants (Mean age = 25.7, SD = 4.3; age range 19–37; 32 females), recruited among students at the University of Oslo, Norway, and forty-two participants (Mean age = 21.3, SD = 2.3; age range 19–32; 23 females), recruited among students at Senshu University, Tokyo, Japan. A *chi-squared* test confirmed that the proportion of student of each gender did not differ ($\chi^2$ = .04, df = 1, p = .84). All participants performed two blocks of the experiment, each lasting approximately 20 minutes. The ethical research boards at the departments of psychology in Oslo (Ref. number 3587571) and in Tokyo (Ref. number 16-S001-1) approved independently the project. All participants signed a written consent form according to the Declaration of Helsinki.

### Stimulus generation

Forty images of female models' faces and forty images of male models' faces were selected among the ~200 faces of young (18–30 years) men and women of the Oslo Face Database, (https://sirileknes.com/oslo-face-database/ [72]). We selected only models' images with a frontal view and direction of gaze for the present study.

Specifically, all 200 faces were divided into four quartiles, for each sex, based on "normative scores" collected independently with 80 participants (40 females) at the University of Oslo; age range = 18–45) who viewed each face separately in random order. The ratings of the male and female faces did not differ on average between raters of different gender and the mean ratings of the same pictures were in fact positively correlated (female faces: R = 0.6; male faces: R = 0.3), as seen in previous studies [73]. We then selected 10 faces from each quartile and paired two models of the same sex that belonged to two different quartiles (e.g., Q1/Q3, Q2/Q4) in each trial.

Each face image was centered using Photoshop® on a light gray circular background (see Fig 1). This circle filled the frame horizontally and vertically and the 40 outermost pixels blended gradually to white. Each face image had a resolution of 512 x 512 pixels (subtending 16.4 degrees of visual angle vertically and horizontally) and the face subtended about 9 degrees of visual angle horizontally when viewed on screen. Faces were matched in pairs, where one model's face was always more attractive, according to norms, than the other in the pair. The faces were band-pass filtered separately with a second order Butterworth filter, using a custom MATLAB-script (this is available from *Figshare*: doi 10.6084/m9.figshare.8311265). The cutoff frequencies in cycles per face width were (lowest—highest): low frequency (0.45–4), medium-low frequency (11.13–18.76), medium-high frequency (25.83–40.5), and high frequency (47.65–77.04). We denoted frequency by *cycles per face width* (cpf), since these are common in the literature [30]. The range of each higher band approximately doubled from the previous band, thus making the bandwidths equal on a $\log_{10}$ scale. For comparison, the cutoff frequencies in *cycles per degree of visual angle* would correspond to the following: low frequency (0.06–0.55), medium-low frequency (1.52–2.5), medium-high frequency (3.54–5.56), and high

frequency (6.53–10.55). After filtering, the faces appeared side by side, resulting in a 1024 x 512-pixel image, and only at this stage the background gray level of the picture of a face pair was set to be equal to the average of the two faces. Finally, we created central fixation images for each pair of both filtered and broadband faces consisting of a black cross (with a size of 0.5˚) that was centered on a gray blank screen (RGB = 128, 128, 128).

## Procedure and data acquisition

The experiment took place for the European participants in the Cognitive Laboratory at the Department of Psychology at the University of Oslo (Norway) and for the Asian participants at the Experimental Psychology laboratory at Senshu University (Japan). A session lasted approximately 1 hour. The procedure of both Norwegian and Japanese testing sessions was identical.

Each trial started with a central fixation cross presented for 1000 ms, followed by the same image but an invisible AOI triggered the presentation of a face pair (Fig 2) whenever gaze was within the AOI surrounding (2˚) the central cross for at least 1 sec. Each face appeared at to the left and to the right of the midpoint. During the whole experiment, a same face appeared four times. Each face pair remained on screen for 5000 ms, a span of time that was sufficient for participants to give a response within the time of each presentation. Participants pressed the Z-key (left) or M-key (right) on a QWERTY keyboard to indicate which of the two faces they found most attractive.

During the whole experiment, iView X software (SMI; Berlin, Germany) recorded eye positions at a sampling rate of 60 Hz. A 4-point calibration procedure preceded each experimental block. In the first block, participants saw 160 pairs of filtered images of same-sex faces, the first 10 trials showed female pairs and then 10 male pairs, repeating this cycle eight times. In the second block, we presented the broadband version of the same 160 pairs of male and females' faces in the same sequence of the previous block. Except for the form of stimuli (filtered or broadband), the task was identical in both blocks.

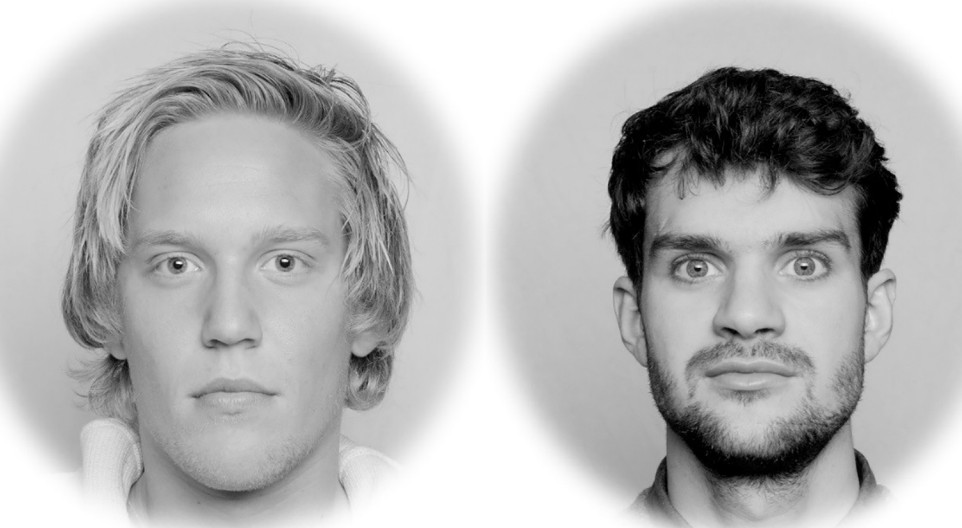

**Fig 2. Illustration of a pair of male faces from the Oslo Face Database in the broadband (unfiltered) condition.** The task in each trial was to decide by pressing one of two keys which of the two faces looked more attractive.

For both the Oslo and Tokyo laboratories, the displays were set at 1280 x 1024 pixels and presented on 47 cm, flat, LCD monitors. Both experimental sessions in Norway and Japan took place in rooms with constant illumination, kept at standard levels throughout testing sessions. Crucially, we collected in both labs the data using equipment of the same model: the Remote Eye-tracking Device 250 (RED), both built in the same year by *SensoMotoric Instruments* (SMI; Berlin, Germany). RED has an automatic compensation for head movements within 0.7 m distance and within a range of 40x20 cm. The Sampling rate was set in both cases at 60 Hz and *BeGaze* software (SMI; Berlin, Germany) used afterwards to detect fixations whenever gaze dwelled for a minimum of 80 ms within a region of maximum 100 pixels, following a standard algorithm.

## Results

### Proportion of choices of the attractive face

We analyzed the averaged data using standard software tools (*StatView* and *JASP*). First, we extracted the proportion of times that each participant selected the same face as most attractive in each of the filtered faces conditions relatively to the broadband (unfiltered) faces condition. Preliminary analyses revealed that Participants' Gender (females, males) did not result in statistical effects and, therefore, we ignored this factor in the following analyses.

We applied a repeated-measure ANOVA to the above 'proportion of times' as the dependent variable and Location (Tokyo or Japan, Oslo or Norway) as between-subjects factors. The within-subject factors were Face Gender (female, male) and Spatial Frequency (SF 1: low-pass filtered face; SF 2: medium-low-pass filtered face; SF 3: medium-high-pass filtered face; SF 4: and high-pass filtered face).

As shown in Fig 3, the mean proportions of times participants groups selected the same face as the most attractive were highest for the medium-low-pass filtered faces (SF 2). This condition differed significantly from the low-pass filtered face (SF 1) and medium-high-pass filtered face (SF 3) but not from the high-pass filtered faces (SF 4). These effects were confirmed by a strong main effect of Spatial Frequency, $F(3, 273) = 48.2$, $p < .0001$, $\eta^2_p = .35$, and a significant Spatial Frequency by Face Gender interaction, $F(3, 273) = 5.6$, $p < .001$, $\eta^2_p = .02$.

While the 'mean proportions' for male faces were comparable for both the Norwegian (Oslo) and Japanese (Tokyo) participants, the Norwegians showed higher correspondence (Mean Proportion = .79) between broadband and filtered conditions for female faces than the Japanese participants did (Mean Proportion = .70). These effects were confirmed by a main effect of Location, $F(1, 91) = 16.2$, $p < .0001$, $\eta^2_p = .15$, which also interacted with Face Gender, $F(1, 91) = 15.5$, $p < .0001$, $\eta^2_p = .12$. There was a slightly higher correspondence between filtered and broadband choices for the Norwegian than the Japanese group in the high SF bands 3 and 4 (Oslo: Mean Proportions = .77; and .80; Tokyo: Mean Proportions = .69 and .74) as seen by the interactive effect of Location by Spatial Frequency, $F(3, 273) = 2.8$, $p = .04$, $\eta^2_p = .03$. There were no other significant main effects or interactions.

### Dwell time of gaze on faces of different attractiveness

To assess differences in dwell times of gaze, we created two Areas of Interest (AOI) corresponding to a) the 'attractive' face in each pair, according to norms, and b) the relatively 'unattractive' face in the same pair. We then computed the mean percentage dwell time spent within each AOI, for each participant, by use of *BeGaze* software (SMI®). We then performed a repeated-measure ANOVA with Location (Tokyo or Japan, Oslo or Norway) as between-subjects factor. The within-subject factors were Choice (attractive, unattractive), Face Gender

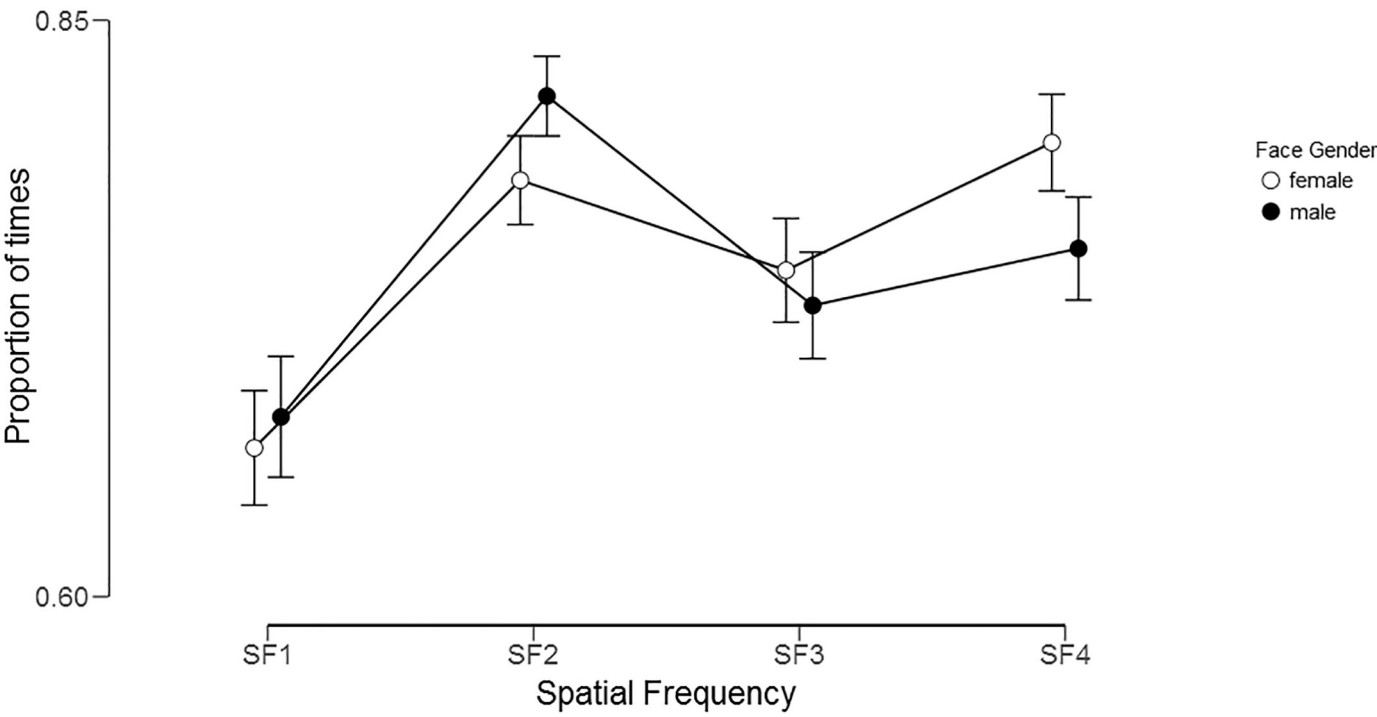

**Fig 3. Proportion of times participants groups selected the same face as the most attractive of a pair in both the unfiltered (broadband) faces condition and in a filtered faces condition, split by Face Gender (black circles = male faces; white circles = female faces).** SF 1 = low-pass filtered face; SF 2 = medium-low-pass filtered face; SF 3 = medium-high-pass filtered face; and SF 4 = high-pass filtered face. Error bars represent the 95% confidence intervals.

(female, male) and Spatial Frequency (SF 1: low-pass filtered face; SF 2: medium-low-pass; SF 3: medium-high-pass; and SF 4: high-pass).

Dwell time of gaze on the attractive female faces increased with spatial frequency (Fig 4, top panel), whereas dwell time peaked for the attractive male faces at SF 2 and for the relatively unattractive male faces at SF 3 (see Fig 5, bottom panel). There was a large main effect of Choice, $F(1, 92) = 272.0$, $p < .0001$, $\eta^2_p = .75$, and Choice interacted with Face Gender, $F(1, 92) = 13.1$, $p < .0001$, $\eta^2_p = .13$, and with Spatial Frequency, $F(3, 276) = 29.4$, $p < .0001$, $\eta^2_p = .24$, as well as with Face Gender and Spatial Frequency, $F(3, 276) = 11.2$, $p < .0001$, $\eta^2_p = .11$. Again, there was a strong main effect of Spatial Frequency, $F(3, 276) = 64.6$, $p < .0001$, $\eta^2_p = .41$.

Dwell times on the attractive face were similar for the male face in both groups. However, Norwegians spent more time on the attractive female face than the Japanese did; as confirmed by an effect of Location, $F(1, 92) = 4.6$, $p = .05$, $\eta^2_p = .05$, and a strong interaction between Location and Face Gender, $F(1, 92) = 60.4$, $p < .0001$, $\eta^2_p = .40$.

### Dwell time of gaze on faces of different gender and by participants' group

To reveal how our Japanese and Norwegian participants scrutinized the SF-filtered faces, we extracted the gaze data of each participant for each face and condition. We collected the obtained fixations in a single map, by taking the mean of the fixations for all occurrences of a specific face. This yielded one gaze map, with average dwell time for each of the faces and broadband/filtered conditions. Then, we warped each gaze map for each face to match an average face using the MATLAB function *imwarp*. For each pixel in the desired output image, the corresponding point in the input image was found (not necessarily an integer coordinate), and the nearest pixels were interpolated to produce the value of the output pixel. The geometry of

**Face Gender: Female Face**

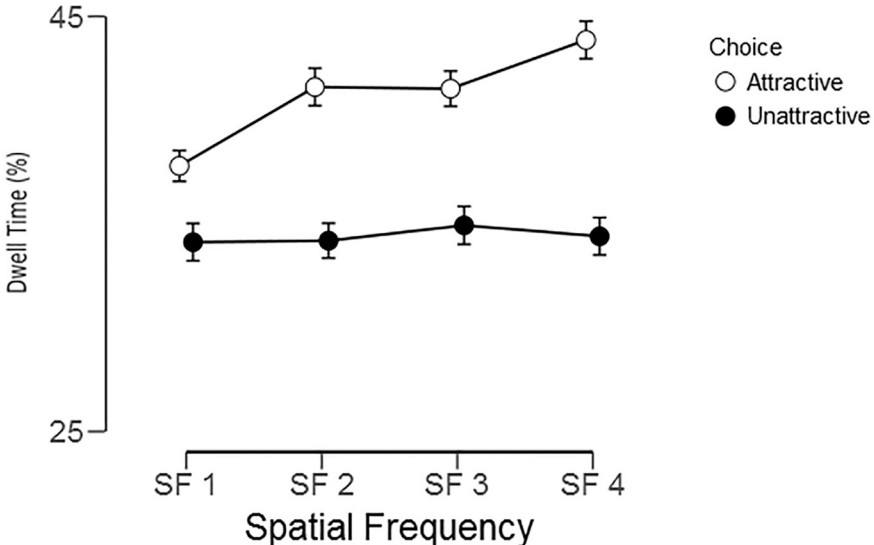

**Face Gender: Male Face**

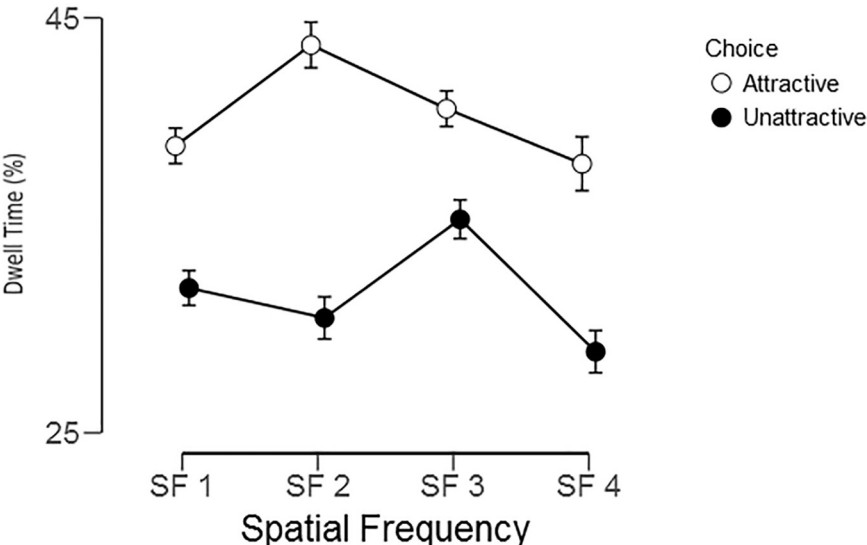

**Fig 4. Percentage dwell times of gaze over the attractive versus unattractive face in each pair, split by filtered faces condition, split by Face Gender (top panel = female faces; bottom panel = male faces).** Error bars represent the 95% confidence intervals.

the output image was determined by applying a geometric transformation of 132 facial landmark points to the average of the facial landmark points for all faces. For each face, we identified 68 facial landmarks based on the Dlib-library [74], and added 68 points relative to the Dlib-landmarks to increase the precision of the warping (Fig 5).

In order to perform statistical analyses on the distribution of gaze, we calculated the percent total dwell time for facial parts. We delineated regions of interest on the average face (Fig 5)

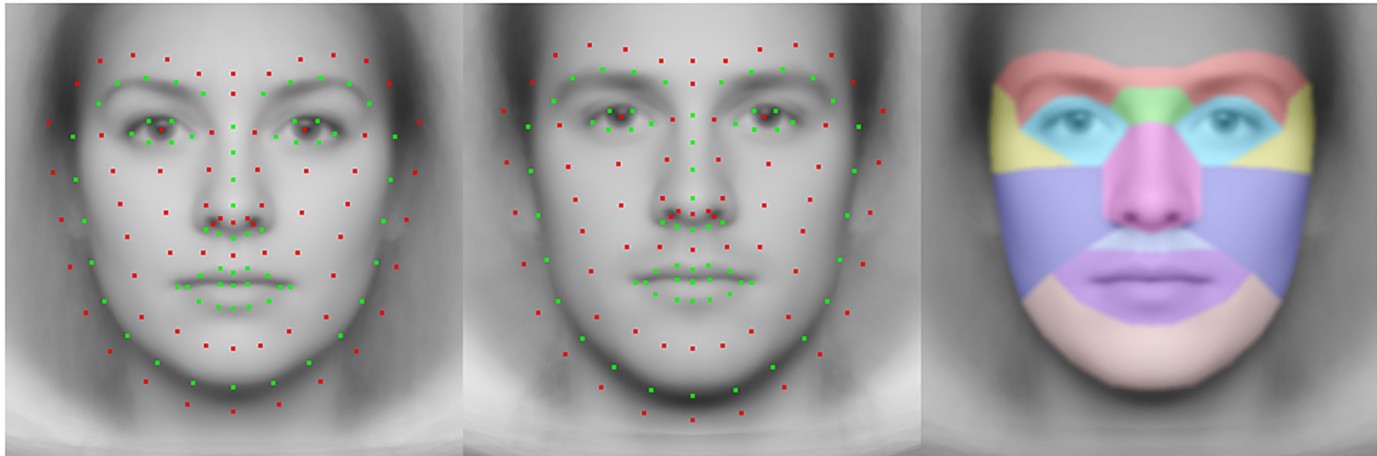

**Fig 5.** *Left and middle panels*: Average of the 132 facial landmarks on the morphs of all female and male faces from the Oslo Face Database. The green points are the landmarks defined by the Dlib-library, and the red points are the added landmarks. The outermost red points help to avoid distortions near the outline of the face following image warping. *Right panel*: The regions (in color) used to calculate the percent dwell times on facial parts are shown superimposed on the androgynous face morph of all faces from the Oslo Face Database.

and the cumulative data were entered in a repeated-measure ANOVA with Location (Oslo, Tokyo) as the between-subjects factor. The within-subject factors were Face Parts (Brows, Nasal Root, Eyes, Temple, Nose, Cheek, Upper Lip, Mouth, Chin), Face Gender (female, male) and Spatial Frequency (SF 1: low-pass; SF 2: medium-low-pass; SF 3: medium-high-pass; and SF 4: high-pass filtered face).

As illustrated in Fig 6, which shows means of dwell time of gaze when viewing the filtered faces, participants fixated the nose most of the time. Such fixations on the nose decreased with increasing SF, whereas fixations on the eyes, which were the second most focused part of the face, increased with increasing SF and more for the female than the male faces (p = .01). Observers directed gaze more to the brows and the nasal root in the LSF than HSF faces, but other face regions did not show much SF dependent changes in dwell time.

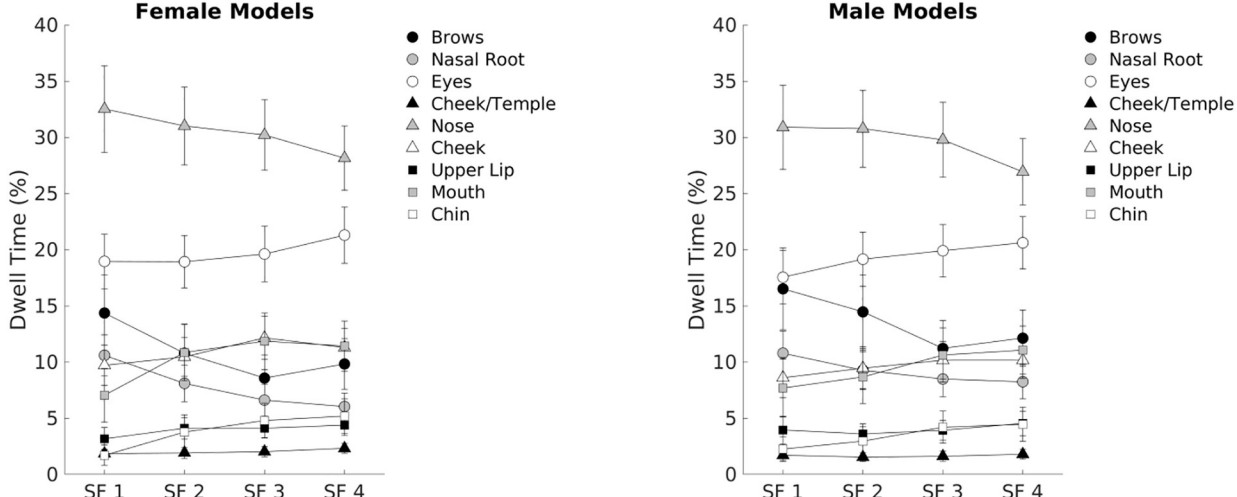

**Fig 6. Percentage dwell times of gaze within each face parts, split by spatial frequency conditions and Face Gender.** Error bars represent the 95% confidence intervals.

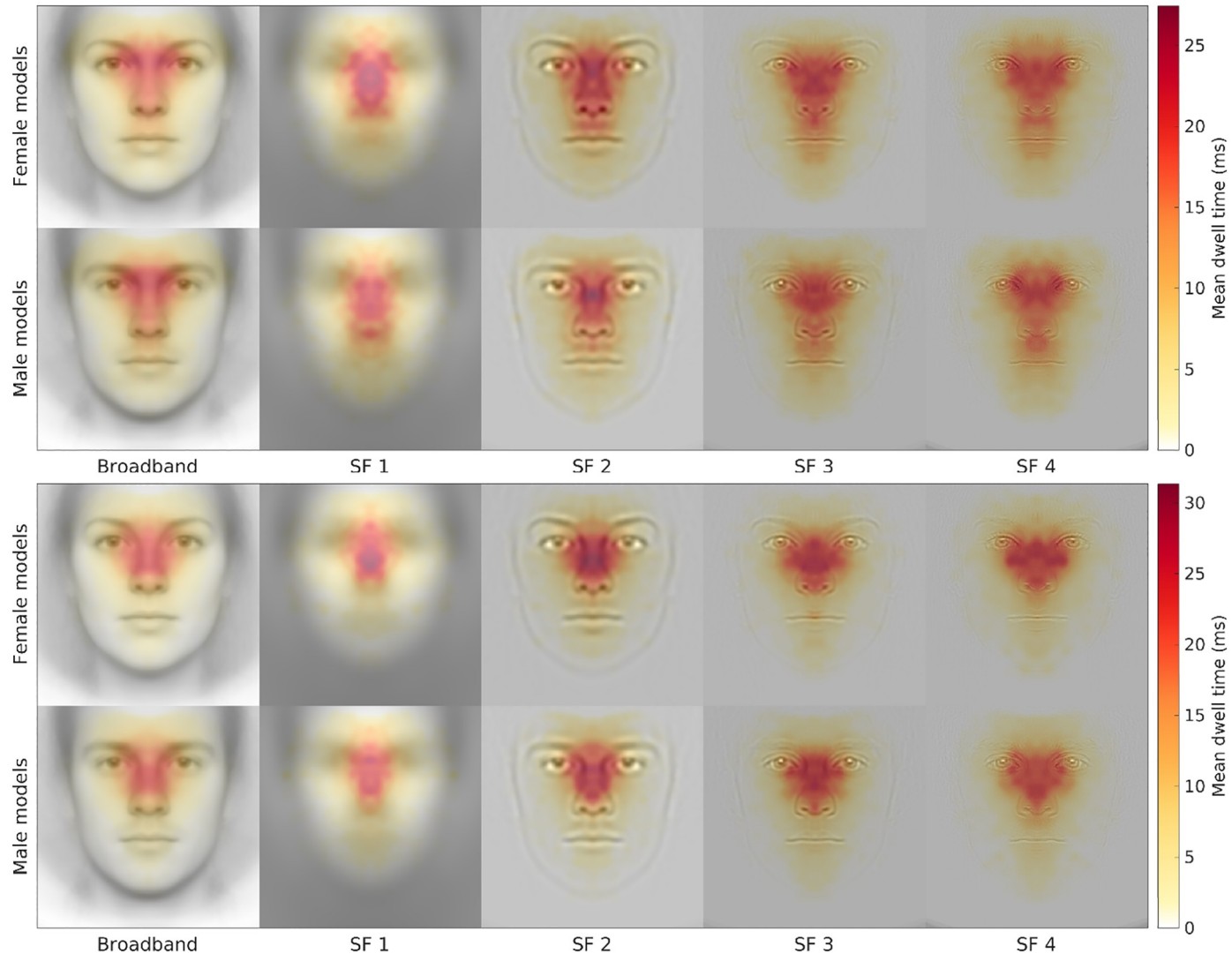

**Fig 7. Mean dwell times of gaze (temperature scale indicates ms) between looking at female versus male models superimposed to the morphs generated from the Oslo Face Database, split by Location (Top panel: Oslo; Bottom panel: Tokyo) and the spatial frequency conditions.**

In Fig 7, we illustrate the interactive effect of Location with Spatial Frequency and Face Gender, expressed as mean dwell time 'attention maps', showing significant differences in rates of fixation on the nose (more for the Japanese), upper lip and mouth (more for the Norwegians).

### Analyses of the structural properties of stimuli

In this section of the results, we present detailed analyses of the stimuli, based on visual properties that are contained within the photo images. We also relate differing structural properties of the female and male faces and in the four spatial frequency bands to the normative attractiveness ratings, as well as to the ratings obtained from Oslo and Tokyo during the corresponding eye-tracking measurements.

In order to assess informational differences between the broadband face and the filtered faces, we formally estimated their structural similarity in female and male models. We used a

variant of the Jensen-Shannon divergence (JSD), which is an estimate of how much two probability distributions have in common [75] or of how informative one distribution is about another distribution. The square root of JSD is a metric [76] or a distance measure. We used $1 - \sqrt{JSD}$ as a similarity measure, and compared the probability distributions of the pixel intensities. The values of the measure lie in the range 0–1, being 1 only if the distributions are identical. Hence, we estimated the similarity between the broadband faces and the filtered faces by measuring the similarity between circular image patches centered on each pixel in the face. The radii of the patches increased from 2 to 32 pixels, in steps of 2, and the resulting measure for a single pixel was the average similarity of all patches centered on that pixel. Thus, for each face, we have a similarity map spanning the entire face. To make the similarity maps for female and male models directly comparable, we warped each similarity map to the average of all faces, as previously described (Fig 6). Before warping, each similarity map was made horizontally symmetrical by first warping the map and the horizontally flipped map to the average of both, then taking the average of the two. This should also reduce the effect of potential differences in lighting between the faces.

We used a non-parametric permutation approach, with 'threshold free cluster enhancement' (TFCE) to compare female and male models. TFCE takes the absolute statistical map and weighs each value based on all lower neighboring values until the local minima [77–78], so that smaller clusters with higher values become comparable with larger clusters with smaller values. We used standard values for the parameters E = 0.5 and H = 2. Permutation testing provides strong control of the family-wise error rate, reducing the risk of false discovery when performing statistical tests on a large number of variables [79]. We ran 5000 permutations to build a null hypothesis distribution. For each permutation, we shuffled the similarity maps between the groups (female/male) before performing $t$-tests with TFCE, and the maximum TFCE value was stored. Finally, we performed $t$-tests on the female versus male similarity maps and thresholded the values according to the null hypothesis distribution of maximum random TFCE values to determine significant areas. The analysis revealed eight significant clusters (see Fig 8). Because of the horizontal symmetry of the similarity maps, the mirrored cluster pairs are identical and we regard each as one cluster.

For the low spatial frequencies, we found four significant clusters of high similarity: 1) in the area around the brows, nasal root, and eyes, for male faces (peak $t$ = -5.93, peak $p < .0001$); 2) in the area around the nose and cheeks for female faces (peak $t$ = 5.64, peak $p < .0001$); 3) around the upper lip and mouth for female faces (peak $t$ = 4.09, peak $p < .0001$); 4) in the cheekbone for male faces (peak $t$ = -4.47, peak $p < .0001$). For the medium low spatial

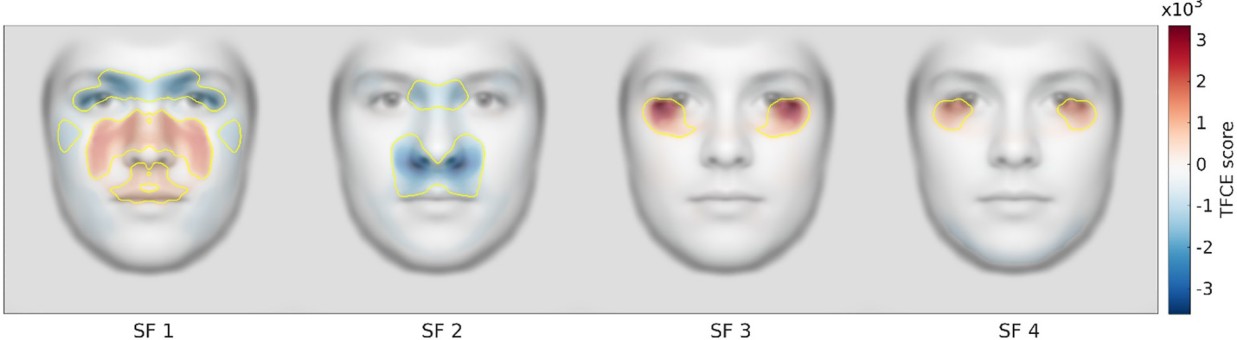

**Fig 8. Threshold free cluster enhanced t-scores for the similarity between the broadband face and the four frequency bands for female versus male models (Red: female > male. Blue: male > female).** Statistically significant clusters are outlined in yellow.

frequencies, the similarity was high for male faces in the nasal root (peak $t$ = -5.53, peak $p <$ .0001) and the area around the nostrils, nose wings, and upper lip (peak $t$ = -7.34, peak $p <$ .0001). For the medium high and high spatial frequencies, the similarity was high for female faces in the area around the lower eyelid and infraorbital region (peak $t$ = 7.61, peak $p <$ .0001; peak $t$ = 6.67, peak $p <$ .0001).

We then estimated the Lempel-Ziv complexity for the significant clusters. This is a measure of Kolmogorov complexity [80], which is the length of the shortest computer program that will reproduce the data, and it can be considered also as a measure of the amount of information contained in the data (see Fig 9). We performed $t$-tests with Holm-Bonferroni correction to compare female and male models. For the low spatial frequencies, the area around the brows, nasal root and eyes contained more information for male models than for female models ($t$(39) = -3.38, $p$ = .002). For medium low spatial frequencies, the nasal root contained more information for male models than for female models ($t$(39) = -5.38, $p <$ .0001), and the area around the nostrils, nose wings, and upper lip contained more information for male models than for female models ($t$(39) = -7.78, $p <$ .0001). For medium high and high spatial frequencies, the area around the lower eyelid and infraorbital region contained more information for female models than for male models ($t$(39) = 4.93, $p <$ .0001; $t$(39) = 6.18, $p <$ .0001).

We then used the European normative data, according to a Likert scale (since the present experimental task used only a forced choice response), to assess which spatial frequencies were related to attractiveness ratings. We estimated for each face image the amplitude of 100 frequencies (logarithmically spaced, from 2 to 256 cycles per image), with 100 orientations (linear spacing, covering 180 degrees). Then to estimate the amplitude, were generated Gabor wavelets for each of the frequencies/orientations and their Fourier transforms were used as frequency filters. The Fourier transform of a Gabor wavelet with a given frequency/orientation is an elliptical Gaussian that peaks at the frequency/orientation in the frequency plane, with its

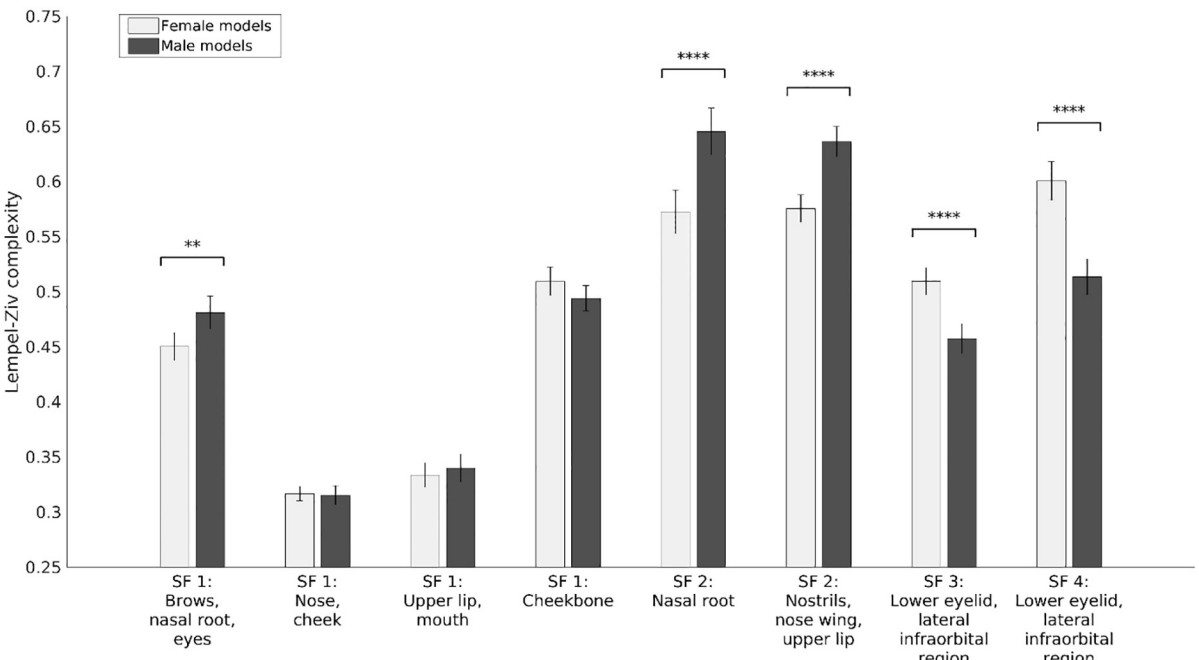

**Fig 9. Lempel-Ziv complexity for female and male models for the significant areas from the analysis of similarity.** Error bars represent 95% confidence intervals. ** p < .01. **** p < .0001.

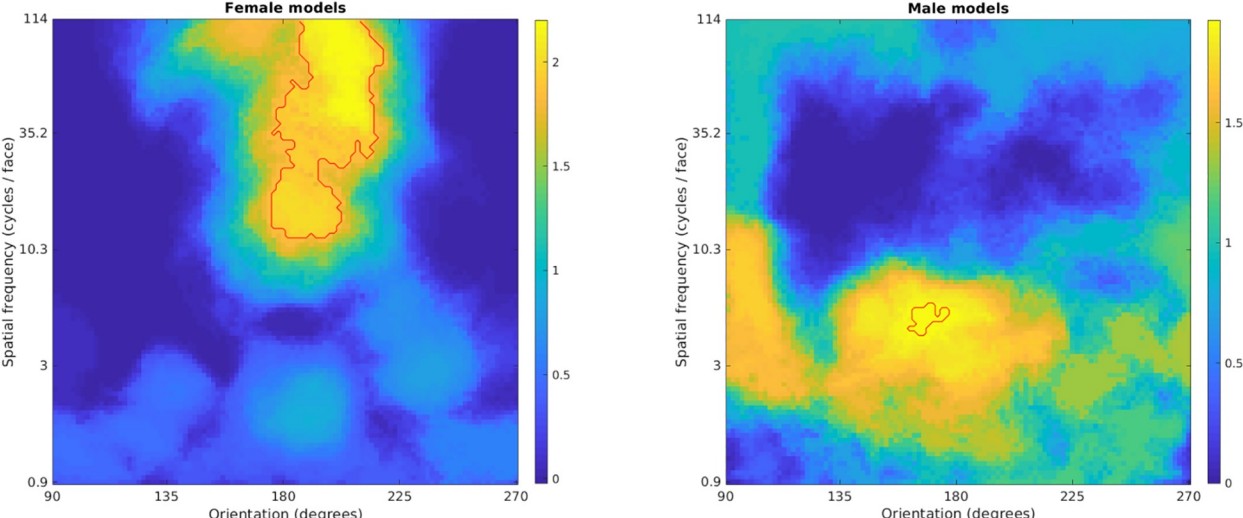

**Fig 10. TFCE-values for correlations between the normative attractiveness ratings and frequency spectra.** Statistically significant clusters are outlined in red. The 90/270 degrees represent horizontal frequencies and 180 degrees the vertical frequencies. There were significant correlations between attractiveness and amplitude for both female faces (at high vertical frequencies) and male faces (at low vertical frequencies).

major axis in the frequency direction. We multiplied each of these with the Fast Fourier Transform amplitude spectrum of the face and took the sum of the product, giving a spectrum with 100x100 frequency bins. After that, we correlated the spectra with the normative ratings, with TFCE and 5000 permutations. For female faces, there was a significant positive correlation between amplitude and attractiveness for approximately vertically oriented frequencies (between 12–1114 cpf). For male faces, there was a significant positive correlation between amplitude and attractiveness for approximately vertically oriented frequencies between 4–6 cpf. Fig 10 illustrates these findings in relation to the normative attractiveness ratings.

Finally, we created graphic visualizations of the female and male face (Fig 11) displaying only the frequencies that were significantly correlated with the normative attractiveness ratings, based on the means of all female and male faces. For each face, we used the significant frequencies as filters, flipping the resulting face horizontally and then taking the mean, warping each to the mean of all faces of the same gender.

A visual inspection of these graphic representations in Fig 11 reveals how attractiveness ratings mostly related for both female and male faces to internal features of the face like the eyebrows, eyes, mouth, as well as the lower part of the face contour or chin, and their immediately surrounding facial surface regions. Note that for the female faces the eye region (i.e., eyelids, irises, pupils) as well as of the lower portion of the nose (nostrils) and the fullness of the lips are clearly visible within these attractiveness-correlated spatial frequencies. Interestingly, female faces' attractiveness-related SFs appear to convey subtle variations over the face surface, mainly in relation to the soft elements of the face and less of its skull structure. In contrast, for the male faces, most of the above facial internal features reveal a rather coarse resolution within the male attractiveness-related SFs and the overall size or extent of the face contour and skull's bone structure dominates the image. Note also that, for the male image, three-dimensional aspects of the whole face or skull structure have more "depth" than for the female face, since the male facial front appears more pronounced than the female's. Indeed, for males, a region above the eyes, including the eyebrows and the bony area immediately above (i.e., the supraorbital process or brow ridge and glabella) as well as a region below the eyes and cheeks' zygomas appear well delineated in their volumetry at these attractiveness-related spatial frequencies.

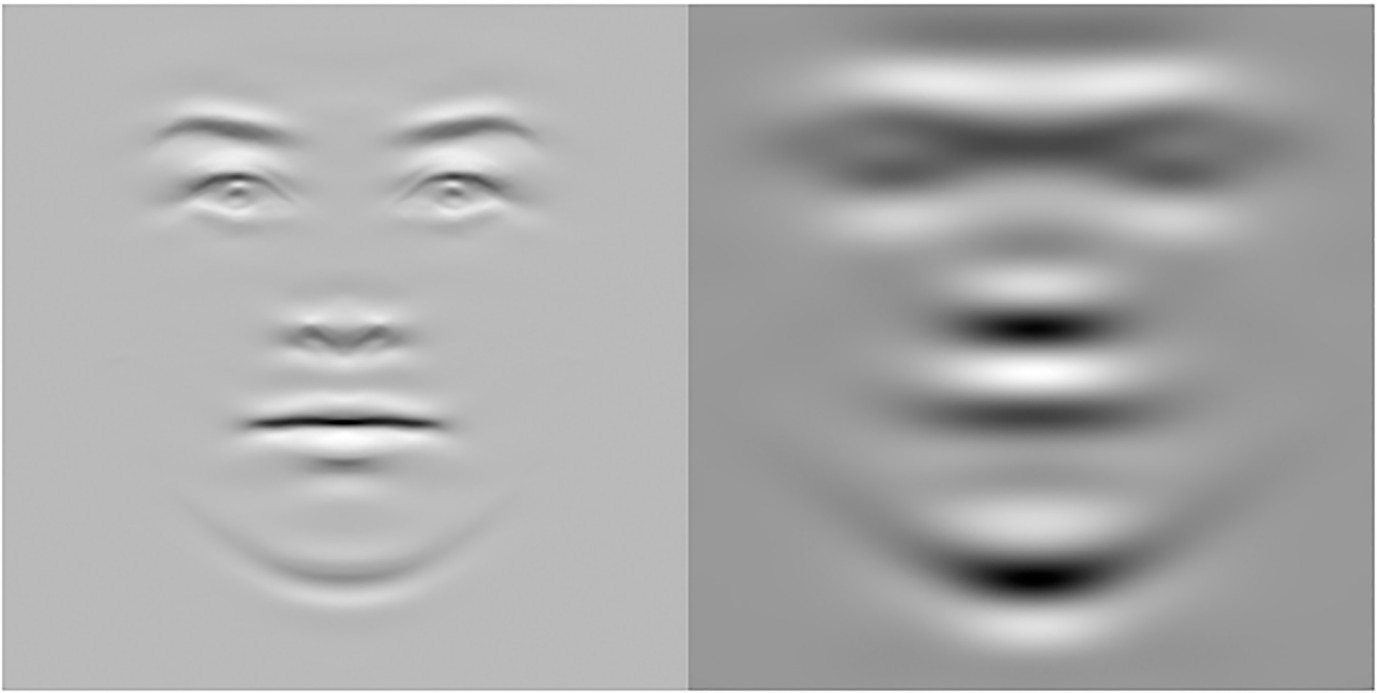

**Fig 11. Graphic representations, based on the means of all female and male faces from the Oslo Face Database, of the spatial frequencies significantly correlated with the normative ratings of attractiveness.**

Moreover, the lower portions of the nose and of the mouth appear to play a relevant role for male attractiveness, although at a coarse level of resolution, so that the separations between the nostrils or lips are not resolved. Finally, the three-dimensional or volumetric aspects of the chin (in particular the protuberance of the jaw or mandible and its breadth) appear to be very salient in the male face image.

## Discussion

A key question about what constitutes our sense of aesthetics is what kind of visual information within the stimulus underlies our judgements. Despite the spatial frequency structure of any visual stimulus is processed very early by the visual brain and several studies have addressed its role in the identification of facial identity and/or expression [81, 82], very few studies have specifically investigated the role of visual spatial frequency information in supporting our sense of facial aesthetics. That different face relevant types of visual information can be optimally channeled through different bands of spatial frequencies is well known for emotional expressions, but the possibility that a similar relationship occur for aesthetic cues has not been fully explored yet. It is very likely that other facial information, seemingly unrelated to visual spatial frequency, plays a relevant role in judgments of attractiveness (e.g., skin's tone [83, 84]), but spatial frequency may play a role beyond the coding of facial shape. In particular, the optimal perception of several of surface and texture cues may be confined within specific bands of spatial information (e.g., the thin lines or creases revealing age or the colors of small parts like the irises). The appreciation of the colors (or discoloring) of small-width or thin facial parts (like the mouth lips) may also depend on high frequency information that may be smeared and significantly weakened in visibility at low spatial frequencies.

Hence, for the present study, we gathered evidence that forced-choice preferences when viewing specific SF bands of face images relate positively to preferences when viewing the corresponding broadband facial images. Our approach consisted in filtering spatial frequencies out of the natural face's photo image (Fig 1). We then presented the obtained face images, containing a narrow band of SF information, in a "beauty contest" between same-sex face pairs (Fig 2). Although all of our photo images depicted faces of really existing European individuals, the participants of the present study belonged to different populations recruited in Europe and Asia (i.e., Norway and Japan).

The gender of the faces had a strong effect on which spatial frequencies were closest related to the same individuals' decisions when performing the task with the unfiltered, natural looking, face pairs. That is, female faces related 80% of the time to choices made with the broadband faces, when viewing the highest of the four SF bands (Fig 3). In contrast, male faces related slightly above 80% of the time to choices made with broadband faces when viewing the second lowest of the four SF bands included in this study (Fig 3). We note that all of the SF bands related above 65% of time to choices made with broadband faces, indicating that all SF bands contribute to some extent with information relevant to aesthetics decision, although apparently in different doses. Thus, it would appear that medium-low spatial frequencies contains visual information that is most relevant for aesthetic decisions made about male faces, but the high spatial frequencies contain key information for decision about female faces. This dissociation is to our knowledge a novel finding, which could lead to identifying detailed gender-specific visual cues.

The oculomotor behavior provided converging evidence for the relevance of the medium-low spatial frequencies for male faces and high spatial frequencies for female faces. The attractive face in a pair was not only looked more in general than the unattractive, but gaze lingered the most over the attractive female face when the face pairs were shown with the highest (SF 4) filtering. Consistently, gaze dwelled the longest over the attractive male face when seeing the face pairs with the medium-low spatial (SF 2) filtering (Fig 4).

In addition, to get a sense of what information is contained and visible in the stimuli and within each band of SF information, we provided visualizations of how this content related to observers' judgements. There was high similarity for the female faces between the two highest (SF 3 and 4) and the broadband faces within a small, bilateral, region (overlapping the eye pupil and the lower eyelid and infraorbital concavity but including the upper part of the zygomatic convexity; Fig 8). In addition, similarity between the filtered and unfiltered faces was higher for male models than female models for the area around the brows, nasal root, and eyes, and especially so for the (medium low) SF 2. Instead, for the medium high and high spatial frequencies (SF 3 and 4), the similarity was higher for female models than male models for the area around the lower eyelid and infraorbital region. Remarkably, the relative distributions of gaze when viewing these SF bands closely matched these similarity profiles.

Statistical analyses on the Lempel-Ziv complexity confirmed that the female faces contained significantly more information than the male faces in the above-described regions (Fig 9). However, information content was significantly high in the low SF bands only for the male faces; in particular, for the central eyebrow region, including the skull area immediately above (i.e., the supraorbital process or brow ridge) in SF 1, and the glabella of the nose and lowest nose region (including the nostrils) in SF 2.

Importantly, there was a striking dissociation between SFs for male and female faces in relation to the relevance of vertically oriented frequencies for attractiveness (normative) ratings. As visible in Fig 10, different spatial frequencies related to the ratings, revealing that for female faces, there was a significant positive correlation between attractiveness and amplitude of high vertical frequencies for female faces and at low vertical frequencies for male faces. Taking into

consideration also the eye-tracking data, participants had a strong tendency to look at the faces along the whole axis of the nose (Fig 7), in particular in the European group, extending as low as the upper lip (philtrum) and the Cupid's bow at the center of the mouth, more so with increasing spatial frequency. This gaze behavior seems consistent with the preponderant role of the central, vertically oriented, features for attractiveness (normative) ratings.

Being the nose at the center of both the vertical and horizontal axes of the 'face' (Nota Bene: below the hairline, not the head), it is presumable that it constitutes an important element to focus gaze when evaluating facial proportions, the configuration and global harmony or symmetry of the face [58]. When spatial frequency is high, the volumetric aspect of the nose, relatively more relevant for the male face (Fig 11, right panel), becomes less visible. The nose is the most sexually dimorphic facial trait in its morphology, being on average disproportionally larger in volume in male than female faces [85–87]. While the visibility of the nose's volume decreases that of its shape and symmetry increases with higher spatial frequencies and the latter features appear more relevant for judging the attractiveness of the female faces (Fig 11, left panel). Since gaze scanning (Fig 7) revealed a strong tendency to focus gaze at the root of the nose, or onto the central portion of the face that may correspond to the limiting size for efficient summation of configurational properties of upright (vertically oriented) face information in a single configurational face template [88]. The eyes, being paired features, horizontally centered together with the vertical prominence of the nose [89], may also convey essential information on a face's proportions and symmetry, and more clearly so in higher spatial frequencies conditions.

It also seems of interest that the dispersion of gaze over the eyes, nose and mouth region differs in our European and Asian groups (Fig 7). The typical T-shaped focus pattern appears mainly with the European participants and increasingly so with higher spatial frequencies. In fact, the pattern of fixations is consistent with previous reports that Asians (i.e., Chinese) tend to look less at the eyes and distribute less their gaze over the face [90–92]. Especially within Japanese culture, a prolonged eye contact may be disrespectful and Japanese children are taught to look at others' necks instead of the eyes [93, 94].

Perhaps the most remarkable dissociation between female and male features related to attractiveness, revealed by the present study's Fourier approach, is between the two faces in Fig 11. These show graphic representations of the spatial frequencies that correlate positively with the stimuli's normative attractiveness ratings (collected independently of the present eye-tracking study and only with Norwegian raters). A striking difference between the two genders' images is that they show very different, little overlapping, SF components. Moreover, these SF components impressively overlap with the SF bands most relevant for forced choices, derived from the present eye-tracking study (Fig 3). For the male face (Fig 11, right panel), the attractiveness-correlated SF provide only a coarse visual resolution of the face, which however clearly conveys the depth or volumetric aspect of the head and face, with its overall size, extent of the face contour (the jaw and chin), and skull's bone structure. These three-dimensional aspects of the male's whole face or skull structure may be important in judging overall proportions. In contrast, the female face's (in Fig 11, left panel) attractiveness-correlated SFs, not only show little overlap with the male's, but they suggest that female attractiveness may be judged more on information carried by higher spatial frequencies. These may reveal local information about the surface of the face and of specific features at a level of detail that is optimal also for the task of individual person recognition and the communication of emotional signals.

In particular, internal features of the female face like the brow ridge, eyes, mouth, as well as the lower part of the face contour or chin, and their immediately surrounding facial surface regions, are clearly visible in the left panel image. We surmise that the high resolution of the above traits allows a more precise evaluation of the arrangements, spatial relations, or distance

ratios between these features (e.g., the inter-ocular distance). There are several suggestions in the literature on facial beauty (also from anthropology, odontology, and aesthetic medical surgery) that our sense of face attractiveness may seek a "golden ratio" between facial traits like the eyes and mouth/teeth and the general proportions of the face ([95] but see [96]). We surmise that at HSF resolutions, information is optimal for spotting the presence of skin blemishes and the smoothness surface skin (i.e., cues of age or poor health) as well as details of the eye region affording the registering of subtle differences in eyelids' and orbital region shape. If smooth skin is crucial for attractiveness in female faces and these properties of surface skin are best represented in high spatial frequencies, then amplitudes of higher frequencies should correlate with attractiveness ratings, since these frequencies make visible these aspects. We also note that the irises' colors as well as the size of the pupils seem clearly delineated at such resolution. Instead, the colors of the irises would be smeared at LSF and, interestingly, previous research suggests that eye color may be more relevant when judging female than male faces for attractiveness [13]. Similarly, the highly mobile pupils may be particularly important for signaling social agreeableness, interest and attraction [8, 97]. We note that our behavioral and gaze results in the main experiment seem consistent with this ideas.

Moreover, the lower portion of the nose (nostrils) and the fullness of the lips (or vermilions) appear clearly visible within these attractiveness-correlated spatial frequencies and shape imperfections and coloring, luminance contrast between sides of the Cupid's bow, may be very salient at this high resolution. Thus, female faces' attractiveness-related SFs may reveal subtle deformations over the face surface, skin, and be related to the soft and malleable elements of the face, instead of its rigid skull structure. These highly mobile parts of the face like the mouth, eyes and eyebrows, all allow the display of subtle affiliative emotions [98], which may also play a key role when judging the attractiveness of an individual, even when just looking at static images [72].

In the male image in Fig 11 (right panel), a region around the ocular orbits, including the eyebrows and the bony area immediately above (i.e., the supraorbital process or ridge and glabella), as well as a region below the eyes and cheeks' zygomas, appears well delineated in volumetry. Interestingly, the lower portions of the nose and of the mouth's upper region play a role for male attractiveness, despite at such a coarse level of resolution the separations between the nostrils or lips are not resolved. Instead, the three-dimensional or volumetric aspects of the chin (in particular the protuberance of the mandible and its breadth) appear to be very salient. A possibility is that the coarse LSF prevalence in the image, by revealing the bony prominence of the brow ridge and of the jaw and chin, conveys effectively the attribute of masculinity inherent in the face [62, 99, 100]. In addition, a large face size characterizes masculinity as opposed to femininity [101]. However, several researchers have cautioned that masculinity may predict attractiveness relatively weakly compared to other fluctuating properties like skin color [102–104] or face and body symmetry [105, 106], which signal immunocompetence. Said and Todorov [18] found a gender-specific dissociation in the effects of shape (e.g., face width) or reflectance (e.g., lightness and color of skin). Increases towards masculinity in reflectance aspects of the male face increased attractiveness, but doing the same in shape aspects decreased it. We surmise that despite the coarse LSF male image (in Fig 11) both the reflectance of skin and of the brows are clearly visible. Interestingly, the reflectance dimensions with the strongest effects on female attractiveness involved the contrast around the eyes and the redness of the lips, which may be both best visible at higher SF.

Indeed, the HSF prominence in the image of the female face' in Fig 11 yields a more detailed but somewhat less volumetric rendition (with slightly "embossed" features to use an art metaphor). What is visible appears related not only to highly mobile parts of the face that allow the display of subtle affiliative emotions but also to several cues associated with a sense of femininity [107, 108]. Sexual dimorphism correspond to different directions in morphometric

space [108] and the female direction is associated with horizontal reduction of the chin, a forward movement of the gonion (jaw angle) and alveolar prognathism. In Fig 11, the male chin is clearly more visible than the female and appears larger in the morphed image.

The eye-tracking results confirmed that the beautiful faces are strong attractors of attention [109], since participants spent about 10% more time dwelling onto the attractive face in a pair (Fig 5) than on the relatively less attractive one. It has been shown that the attentional priority towards attractive faces can also occur unconsciously [110] and that a decision about a face's level of attractiveness can be reached very rapidly (within 33 ms), and not very differently than when having unlimited time [111]. However, the present results are consistent with several previous studies showing that we typically spend extra time looking at faces considered attractive [112–114].

Finally, a previous study [38] used Fourier power spectrum analyses to describe the relation between spatial frequency and power of the radially averaged (1d) Fourier spectrum on a log-log scale. As the researchers point out, most natural (complex) images show a linear relationship and the relative strength or 'power' of fine detail information or coarse structure in an image can be, respectively, expressed linearly be the angle of the slopes in the power plots. Importantly, enhanced HSF information leads to shallow slopes, whereas enhanced LSF information leads to steep slopes. Given that pleasing natural scenes and artworks share a shallow power slope of -2 [115], the authors hypothesized that also faces approaching a Fourier power slope of -2 (i.e., with enhanced HSF information) would be considered more attractive than the same face, or others, differing from this value (e.g., steeper slopes between -3 and -4). Remarkably, when participants were given the opportunity to manually adjust the Fourier slope of the images on screen, they did choose a mean value of -2.6, which is a bit closer to that of pleasing natural scenes or artistic facial portraits. The effect was significantly larger for female faces, which also seems consistent with the present study's findings of a bias for HSF information for female faces. A limitation of the Fourier slope approach is that it is informative about the relative distribution of frequency power, but not specific frequency bands. We surmise that, by presenting ranges of SF information separately, we are likely to reveal which information contained in the natural stimulus directly related to the aesthetic judgment about a face. In contrast, by strengthening or adding one type *of v*isual information by distorting the natural image, one can reveal directional biases and explore the limits within which a face's attractiveness can be enhanced [17].

## Conclusions

Data collected in both locations (Oslo and Tokyo) revealed that the highest mean proportion of preference for the most attractive face in a pair occurred with the highest frequencies band for female faces and the medium-low frequency band for male faces. Thus, the present spatial frequency analysis leads to the conclusion that the most relevant information for judging male faces' attractiveness is within the low spatial frequencies, while the most relevant information for deciding upon female faces' beauty within the high spatial frequencies. These SF related findings provide converging evidence for the relevance of specific facial traits to the sense of attractiveness for each gender and they hint to a greater relevance of the mobile and communicative parts of the face for the female face and of the rigid, structural, parts for the male face.

## Author Contributions

**Conceptualization:** Morten Øvervoll, Matia Okubo, Bruno Laeng.

**Data curation:** Morten Øvervoll, Ilaria Schettino, Hikaru Suzuki.

**Formal analysis:** Morten Øvervoll, Bruno Laeng.

**Investigation:** Ilaria Schettino, Hikaru Suzuki, Bruno Laeng.

**Methodology:** Morten Øvervoll, Matia Okubo, Bruno Laeng.

**Project administration:** Morten Øvervoll, Bruno Laeng.

**Supervision:** Morten Øvervoll, Matia Okubo, Bruno Laeng.

**Visualization:** Morten Øvervoll.

**Writing – original draft:** Morten Øvervoll, Bruno Laeng.

**Writing – review & editing:** Ilaria Schettino, Matia Okubo.

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
