## [Decision Letter · Decision Letter 0]

2 Oct 2019

PONE-D-19-19674

Filtered beauty in Oslo and Tokyo: A spatial frequency analysis of facial attractiveness

PLOS ONE

Dear Dr. Laeng,

Thank you for submitting your manuscript to PLOS ONE. After careful consideration, we feel that it has merit but does not fully meet PLOS ONE’s publication criteria as it currently stands. Therefore, we invite you to submit a revised version of the manuscript that addresses the points raised during the review process.

Both reviewers (Alexis Makin and Gregor U. Hayn-Leichsenring) commented favourably on the strength of the manuscript and found your study interesting and well-motivated. However, the reviewers also shared concerns mainly regarding the readability of the manuscript and its length, particularly with regard to the number of analyses and figures. In the revision, if you could focus on making the manuscript more concise and streamlined, to convey the main message to the readers more easily.

We would appreciate receiving your revised manuscript by Nov 16 2019 11:59PM. To enhance the reproducibility of your results, we recommend that if applicable you deposit your laboratory protocols in protocols.io, where a protocol can be assigned its own identifier (DOI) such that it can be cited independently in the future. For instructions see: http://journals.plos.org/plosone/s/submission-guidelines#loc-laboratory-protocols

We look forward to receiving your revised manuscript.

Kind regards,

Magdalena Ewa Król, Ph.D.

Academic Editor

PLOS ONE

Journal Requirements:

2.  We note that Figure(s) in your submission contain copyrighted images. All PLOS content is published under the Creative Commons Attribution License (CC BY 4.0), which means that the manuscript, images, and Supporting Information files will be freely available online, and any third party is permitted to access, download, copy, distribute, and use these materials in any way, even commercially, with proper attribution. For more information, see our copyright guidelines: http://journals.plos.org/plosone/s/licenses-and-copyright.

1.    You may seek permission from the original copyright holder of Figure(s) [#] to publish the content specifically under the CC BY 4.0 license.

3. We note that Figures in your manuscript include images of a [patient / participant / in the study]. 

If you are unable to obtain consent from the subject of the photographs, you will need to remove the figure and any other textual identifying information or case descriptions for this individual.

Additional Editor Comments (if provided):

Reviewers' comments:

Reviewer's Responses to Questions

**Comments to the Author**

1. Is the manuscript technically sound, and do the data support the conclusions?

Reviewer #1: Yes

Reviewer #2: Yes

2. Has the statistical analysis been performed appropriately and rigorously? 

Reviewer #1: Yes

Reviewer #2: Yes

3. Have the authors made all data underlying the findings in their manuscript fully available?

Reviewer #1: Yes

Reviewer #2: Yes

4. Is the manuscript presented in an intelligible fashion and written in standard English?

Reviewer #1: Yes

Reviewer #2: Yes

5. Review Comments to the Author

Reviewer #1: This is an interesting study, both novel and well-motivated. The authors ask whether different spatial frequencies carry special information about the facial beauty. Do the important frequencies depend on the sex of the face? Are effects universal or culturally relative? The most impressive results were that low spatial frequencies gave a good approximation of broadband attractiveness ratings in male faces, but high spatial frequencies gave a good approximation of broadband attractiveness ratings in female faces.

This dissociation was paralleled by measures of dwell time from additional eye tracking analysis.

I recommend the study should be published.

My main recommendation is to streamline the manuscript and make it more digestible. At the moment, I feel very few readers will have the patience to work through all 17 figures and all the involved analysis, and extract the key points.

I feel the authors can make better judgements than me about how do this necessary word-pruning operation. I would want to force the authors to remove any particular analysis or figure, just carefully consider any possible opportunities for ruthless streamlining!

Just because a sophisticated analysis and figure is possible, it does not mean it always helps the reader understand the take home message of the paper. Which sections of the results, and which figures, make an absolutely essential contribution?

Are there cases where two different analyses do the same job? Could you present just one of them, and say briefly, ‘this was backed up by further analysis of X'?

Some of the image control analysis could perhaps be mentioned in the method, rather than cluttering the results section?

I think reducing the number of Figures would help. Could some of these be collapsed into single, multi-panel figures?

Why do Figure 4a and Figure 4b have to be uploaded as separate files? Couldn’t they be combined into one multi-panel figure 4? I suspect this is what the authors already had in mind, but there are other figures, which are currently separate, that could be combined.

I wonder if the results sections would be easier to read if you describe the results before the ANOVAs? As it is, we lead in with the ANOVAs. When the reader is at the end of the ANOVA sentence, they don’t know much, because they don’t know the direction of the effects. For example you say:

“There was also a moderate effect of Location, F(1, 92)= 4.6, p= .05, η2p= .05, and a strong

interaction between Location and Face Gender, F(1, 92)= 60.4, p< .0001, η2p= .40. Whereas dwell time on the attractive face was similar for the male face in both groups, the Norwegians spent more time on the attractive female face than the Japanese did. “

This is not 'wrong', but would I find it much easier if you start with the description. For instance, this example could be reworded:

Whereas dwell time on the attractive face was similar for the male face in both groups, the Norwegians spent more time on the attractive female face than the Japanese did. There was confirmed by a moderate effect of Location, F(1, 92)= 4.6, p= .05, η2p= .05, and a strong

interaction between Location and Face Gender, F(1, 92)= 60.4, p< .0001, η2p= .40.

Finally, there are maybe sections of the discussion which are repetitive or speculative. I would suggest re-reading this with an eye to making it more streamlined. Again, I trust the authors to judge which parts of the discussion are absolutely essential.

Reviewer #2: In their manuscript entitled „Filtered beauty in Oslo and Tokyo: 9 A spatial frequency analysis of facial attractiveness“, the authors describe the effect of spatial frequencies on attractiveness ratings on human faces.

The studies are overall very well conducted, the research question is clear and the results contribute largely to a better understanding of the role of spatial frequencies in judgments on face attractiveness.

Therefore, I recommend a minor revision.

I still have some points that should be addressed before publication.

Major points

l. 650ff, 702 ff and 758ff: I was wondering whether this can be quantified in more detail. Possibly, one could argue that male attractiveness is more related to typical male physiognomy (like wide chin, dark eyebrows, etc) that is represented in lower SFs, while stereotypical female attractiveness is more dependent on a smooth skin (and skin blemishes are more represented in higher SFs). Would you agree? I would expect that lower amounts of high SFs contribute to a higher attractiveness rating in female faces, since lower amounts of high SFs indicate an absence of these blemishes and relate to a smooth skin.

l. 908-910: I disagree (see above). In my understanding, the key for facial attractiveness in male faces lies in the „configuration" (?) of low SFs, but not in the amount of low SFs. I would not expect male faces with a higher amount of low SFs to be rated as more attractive, since the amount is not a measure for the facial shade that is represented (at least partly) in low SFs. This should not be the case for higher amounts of low SFs contributing to higher attractiveness ratings in male faces, since higher amounts of low SFs do not indicate a different facial structure. Is there any possibility to check for this sub-hypothesis?

The authors seem to switch inconsistently between the terms „beauty“ and „attractiveness“, I strongly suggest to be consistent within the manuscript (personally, I prefer the term facial attractiveness).

One of my biggest concerns is the length of the manuscript. To me, the inclusion of the eye-tracking part, although interesting, distracts from the - in my opinion - behavioral main findings. I’d prefer a shorter manuscript with less data but one consistent message. The eye-tracking results might enough for an additional manuscript. However, if the authors want to include the eye-tracking results, I am ok with it.

Minor points:

l. 247: „ratings across pictures“

l. 262: reference?

l. 321: typo, I guess it should state „SF 3“

Figures 3 and 4A+B look strange, might be an image compression problem.

Figure 5: legend for y-axis is missing

l. 412: was this effect significant?

l. 425: please indicate p-values for ** and ***

l. 437: just wanted to note that I like this chapter. There is a lot of important information provided.

l. 724: „considered“

l. 859ff: I am not sure whether this paragraph is necessary for the story. I’d suggest eliminating it from the manuscript since dominance and trustworthiness (although important characteristics) are not mentioned anywhere else in the manuscript.

l. 899: „-2“, not „-.2“

Is it possible to structure the discussion? 2nd degree headlines would be great to help the reader finding the part of the discussion she/he is interested in.

6. PLOS authors have the option to publish the peer review history of their article (what does this mean?). If published, this will include your full peer review and any attached files.

Reviewer #1: Yes: Alexis Makin

Reviewer #2: Yes: Gregor U Hayn-Leichsenring

---

## [Author Response · Author response to Decision Letter 0]

6 Nov 2019

Reviewer #1: 

This is an interesting study, both novel and well-motivated. The authors ask whether different spatial frequencies carry special information about the facial beauty. Do the important frequencies depend on the sex of the face? Are effects universal or culturally relative? The most impressive results were that low spatial frequencies gave a good approximation of broadband attractiveness ratings in male faces, but high spatial frequencies gave a good approximation of broadband attractiveness ratings in female faces. This dissociation was paralleled by measures of dwell time from additional eye tracking analysis. I recommend the study should be published.

Response: We thank the Reviewer for the positive evaluation of our study and, in particular, for finding in it both novelty and sound theoretical grounding. We also appreciate that the Reviewer found value in our parallel measures of dwell time based on eye tracking.

My main recommendation is to streamline the manuscript and make it more digestible. At the moment, I feel very few readers will have the patience to work through all 17 figures and all the involved analysis, and extract the key points. I feel the authors can make better judgements than me about how do this necessary word-pruning operation. I would want to force the authors to remove any particular analysis or figure, just carefully consider any possible opportunities for ruthless streamlining! Just because a sophisticated analysis and figure is possible, it does not mean it always helps the reader understand the take home message of the paper. Which sections of the results, and which figures, make an absolutely essential contribution? Are there cases where two different analyses do the same job? Could you present just one of them, and say briefly, ‘this was backed up by further analysis of X'? Some of the image control analysis could perhaps be mentioned in the method, rather than cluttering the results section?

Response: We appreciate the Reviewer’s suggestion and we have indeed shortened the manuscript considerably. We have removed a whole part of the analyses that we thought would not detract from the main message.

I think reducing the number of Figures would help. Could some of these be collapsed into single, multi-panel figures?

Response: Indeed, we have collapsed some Figures and deleted a few, so that the current number of Figure is 11 instead of 17.

Why do Figure 4a and Figure 4b have to be uploaded as separate files? Couldn’t they be combined into one multi-panel figure 4? I suspect this is what the authors already had in mind, but there are other figures, which are currently separate, that could be combined.

Response: Done.

I wonder if the results sections would be easier to read if you describe the results before the ANOVAs? As it is, we lead in with the ANOVAs. When the reader is at the end of the ANOVA sentence, they don’t know much, because they don’t know the direction of the effects. For example you say: “There was also a moderate effect of Location, F(1, 92)= 4.6, p= .05, η2p= .05, and a strong interaction between Location and Face Gender, F(1, 92)= 60.4, p< .0001, η2p= .40. Whereas dwell time on the attractive face was similar for the male face in both groups, the Norwegians spent more time on the attractive female face than the Japanese did. “This is not 'wrong', but would I find it much easier if you start with the description. For instance, this example could be reworded: Whereas dwell time on the attractive face was similar for the male face in both groups, the Norwegians spent more time on the attractive female face than the Japanese did. There was confirmed by a moderate effect of Location, F(1, 92)= 4.6, p= .05, η2p= .05, and a strong interaction between Location and Face Gender, F(1, 92)= 60.4, p< .0001, η2p= .40.

Response: We have followed the Reviewer’s suggestions and inverted the order of the discursive and numerical descriptions of specific results.

Finally, there are maybe sections of the discussion which are repetitive or speculative. I would suggest re-reading this with an eye to making it more streamlined. Again, I trust the authors to judge which parts of the discussion are absolutely essential.

Response: We have done our best to streamline the Discussion and we hope that the current version has gained in flow and shows no redundancies.

---

Reviewer #2: 

In their manuscript entitled „Filtered beauty in Oslo and Tokyo: 9 A spatial frequency analysis of facial attractiveness“, the authors describe the effect of spatial frequencies on attractiveness ratings on human faces. The studies are overall very well conducted, the research question is clear and the results contribute largely to a better understanding of the role of spatial frequencies in judgments on face attractiveness. Therefore, I recommend a minor revision.

Response: We thank the Reviewer for the positive evaluation of our study and, in particular, for the statement that our ‘results contribute largely to a better understanding of the role of spatial frequencies in judgments on face attractiveness.’

I still have some points that should be addressed before publication. Major points

l. 650ff, 702 ff and 758ff: I was wondering whether this can be quantified in more detail. Possibly, one could argue that male attractiveness is more related to typical male physiognomy (like wide chin, dark eyebrows, etc) that is represented in lower SFs, while stereotypical female attractiveness is more dependent on a smooth skin (and skin blemishes are more represented in higher SFs). Would you agree? I would expect that lower amounts of high SFs contribute to a higher attractiveness rating in female faces, since lower amounts of high SFs indicate an absence of these blemishes and relate to a smooth skin.

l. 908-910: I disagree (see above). In my understanding, the key for facial attractiveness in male faces lies in the „configuration" (?) of low SFs, but not in the amount of low SFs. I would not expect male faces with a higher amount of low SFs to be rated as more attractive, since the amount is not a measure for the facial shade that is represented (at least partly) in low SFs. This should not be the case for higher amounts of low SFs contributing to higher attractiveness ratings in male faces, since higher amounts of low SFs do not indicate a different facial structure. Is there any possibility to check for this sub-hypothesis?

Response: We believe that the Reviewer raises an important issue and we agree that male attractiveness could be most related to typical male physiognomy, which is represented in lower spatial frequencies. Our results appear to support this idea. Moreover, the positive correlation between spatial frequency amplitude and attractiveness ratings suggests that a set of relatively low spatial frequencies (around vertical orientations) are more important for male than female attractiveness and that increased amplitudes in these frequencies are associated with increased attractiveness ratings. That is, the amplitudes of these frequencies indicate the degree in which certain facial features are present or visible in a face (e.g., how prominent is the supraorbital ridge). Nota Bene: We do not suggest that an increased “amount of low spatial frequencies” per se is important – as in the number of frequencies present in an image – but rather that the amplitude of a set of frequencies, or the visibility/presence of certain facial features represented by these frequencies, are important.

We also agree with the Reviewer that smooth skin is crucial for attractiveness in female faces and that skin blemishes and properties of surface skin are best represented in high spatial frequencies. Both our behavioral and gaze results seem consistent with this ideas. If smooth skin was the most important factor for female attractiveness, then amplitudes of higher frequencies should correlate with attractiveness ratings, since these frequencies make visible these aspects. That is, the positive correlation between these frequencies and attractiveness ratings implies that increased amplitude (increased visibility/presence of these facial features) is associated with increased attractiveness ratings. However, we note that other facial features are also represented by high spatial frequencies and our results suggest that a specific set of high spatial frequencies are relevant for attractiveness in female faces. We also do not rule out a possible negative correlation between attractiveness and a specific set of high frequencies that may be closely associated with skin blemishes. However, we have not attempted to disentangle high frequencies in this manner and it is not clear at the present which approach would be most appropriate, but this may be a very interesting point to look into in the future. We hope that the above reflections provide some answers to the Reviewer’s insightful remarks. At any rate, we have added a sentence about skin blemishes and female attractiveness and have removed the paragraph (Lines 908-915) that, in the original manuscript, started with “However, there was no preference for Fourier steep slopes for the male faces…” Thus, in the new manuscript we do not make controversial statements about steepness of slopes in the Fourier approach in relation to male attractiveness.

The authors seem to switch inconsistently between the terms „beauty“ and „attractiveness“, I strongly suggest to be consistent within the manuscript (personally, I prefer the term facial attractiveness).

Response: We have now substituted most instances of the term ‘beauty’ with the term ‘attractiveness’. We have kept ‘beauty’ in the first line of the Title, since ‘facial attractiveness’ is mentioned in the second line.

One of my biggest concerns is the length of the manuscript. To me, the inclusion of the eye-tracking part, although interesting, distracts from the - in my opinion - behavioral main findings. I’d prefer a shorter manuscript with less data but one consistent message. The eye-tracking results might enough for an additional manuscript. However, if the authors want to include the eye-tracking results, I am ok with it.

Response: We have greatly reduced the length of the manuscript (of about one third), since both reviewers commented about this aspect. We have decided however to keep the eye-tracking results since they appear to provide remarkable converging evidence with the other data. We appreciate that the Reviewer’s anticipated such a decision and showed flexibility towards it.

Minor points:

l. 247: „ratings across pictures“

l. 262: reference?

l. 321: typo, I guess it should state „SF 3“

Figures 3 and 4A+B look strange, might be an image compression problem.

Figure 5: legend for y-axis is missing

l. 412: was this effect significant?

l. 425: please indicate p-values for ** and ***

l. 437: just wanted to note that I like this chapter. There is a lot of important information provided.

l. 724: „considered“

l. 859ff: I am not sure whether this paragraph is necessary for the story. I’d suggest eliminating it from the manuscript since dominance and trustworthiness (although important characteristics) are not mentioned anywhere else in the manuscript.

l. 899: „-2“, not „-.2“

Response: We thank the Reviewer for spotting several oversights and typos. We have now corrected these (in those parts of the text that have survived our editing).

Is it possible to structure the discussion? 2nd degree headlines would be great to help the reader finding the part of the discussion she/he is interested in.

Response: We found difficult to organize the current Discussion with headlines and we hope that the present format is acceptable.

---

## [Decision Letter · Decision Letter 1]

20 Dec 2019

Filtered beauty in Oslo and Tokyo: A spatial frequency analysis of facial attractiveness

PONE-D-19-19674R1

Dear Dr. Laeng,

We are pleased to inform you that your manuscript has been judged scientifically suitable for publication and will be formally accepted for publication once it complies with all outstanding technical requirements.

With kind regards,

Magdalena Ewa Król, Ph.D.

Academic Editor

PLOS ONE

Additional Editor Comments (optional):

Reviewers' comments:

Reviewer's Responses to Questions

**Comments to the Author**

1. If the authors have adequately addressed your comments raised in a previous round of review and you feel that this manuscript is now acceptable for publication, you may indicate that here to bypass the “Comments to the Author” section, enter your conflict of interest statement in the “Confidential to Editor” section, and submit your "Accept" recommendation.

Reviewer #1: All comments have been addressed

Reviewer #2: All comments have been addressed

2. Is the manuscript technically sound, and do the data support the conclusions?

Reviewer #1: Yes

Reviewer #2: Yes

3. Has the statistical analysis been performed appropriately and rigorously? 

Reviewer #1: Yes

Reviewer #2: Yes

4. Have the authors made all data underlying the findings in their manuscript fully available?

Reviewer #1: Yes

Reviewer #2: Yes

5. Is the manuscript presented in an intelligible fashion and written in standard English?

Reviewer #1: Yes

Reviewer #2: Yes

6. Review Comments to the Author

Reviewer #1: I can see that the authors have addressed my concerns and improved the manuscript. I was already quite positive about the manuscript, so this is now ready for publication.

Reviewer #2: Thank you for adressing my comments. I will support a publication of the manuscript.

In my opinion, the manuscript benefited to a great extend from the revision. As mentioned in the first round of the review process, the study and the presentation are excellent and the - now - shortened manuscript makes it more approachable.

7. PLOS authors have the option to publish the peer review history of their article (what does this mean?). If published, this will include your full peer review and any attached files.

Reviewer #1: Yes: Alexis Makin

Reviewer #2: Yes: Gregor Uwe Hayn-Leichsenring

---

## [Editor Report · Acceptance letter]

31 Dec 2019

PONE-D-19-19674R1 

Filtered beauty in Oslo and Tokyo: A spatial frequency analysis of facial attractiveness. 

Dear Dr. Laeng:

I am pleased to inform you that your manuscript has been deemed suitable for publication in PLOS ONE. Congratulations! Your manuscript is now with our production department. 

With kind regards,

on behalf of

Dr. Magdalena Ewa Król 

Academic Editor

PLOS ONE